# When Bone Forms Where It Shouldn’t: Heterotopic Ossification in Muscle Injury and Disease

**DOI:** 10.3390/ijms26157516

**Published:** 2025-08-04

**Authors:** Anthony Facchin, Sophie Lemaire, Li Gang Toner, Anteneh Argaw, Jérôme Frenette

**Affiliations:** 1Centre Hospitalier Universitaire de Québec-Université Laval Research Center (CHUQ-CHUL), Axe Neurosciences, Université Laval, Quebec City, QC G1V 4G2, Canada; anthony.facchin.1@crchudequebec.ulaval.ca (A.F.); anteneh.argaw@crchudequebec.ulaval.ca (A.A.); 2School of Rehabilitation Sciences, Faculty of Medicine, Université Laval, Quebec City, QC G1V 0A6, Canada; sophie.lemaire.1@ulaval.ca (S.L.);

**Keywords:** skeletal muscle, heterotopic ossification, macrophage, fibro-adipogenic progenitor cell, neuro-inflammation, lymphocyte, extracellular matrix, metabolism

## Abstract

Heterotopic ossification (HO) refers to the pathological formation of bone in soft tissues, typically following trauma, surgical procedures, or as a result of genetic disorders. Notably, injuries to the central nervous system significantly increase the risk of HO, a condition referred to as neurogenic HO (NHO). This review outlines the cellular and molecular mechanisms driving HO, focusing on the inflammatory response, progenitor cell reprogramming, and current treatment strategies. HO is primarily fuelled by a prolonged and dysregulated inflammatory response, characterized by sustained expression of osteoinductive cytokines secreted by M1 macrophages. These cytokines promote the aberrant differentiation of fibro-adipogenic progenitor cells (FAPs) into osteoblasts, leading to ectopic mineralization. Additional factors such as hypoxia, BMP signalling, and mechanotransduction pathways further contribute to extracellular matrix (ECM) remodelling and osteogenic reprogramming of FAPs. In the context of NHO, neuroendocrine mediators enhance ectopic bone formation by influencing both local inflammation and progenitor cell fate decisions. Current treatment options such as nonsteroidal anti-inflammatory drugs (NSAIDs), radiation therapy, and surgical excision offer limited efficacy and are associated with significant risks. Novel therapeutic strategies targeting inflammation, neuropeptide signalling, and calcium metabolism may offer more effective approaches to preventing or mitigating HO progression.

## 1. Introduction

Skeletal muscle is the most abundant tissue type in the human body, comprising approximately 40% of total body mass [1]. It plays a vital role in essential physiological functions, including locomotion, postural support, protein storage, and metabolic regulation, particularly in maintaining glucose homeostasis [1]. Skeletal muscles are connected to bones via tendons, which are composed of dense connective tissue, facilitating movement. Structurally, skeletal muscle is organized into three layers of connective tissue: the epimysium, which encases the entire muscle; the perimysium, which surrounds bundles of muscle fibres known as fascicles; and the endomysium, which envelops individual muscle fibres.

Skeletal muscle contains different cell types, with muscle fibres (myofibres) being the most abundant. The elongated, multinucleated myofiber spans from one tendon to another. Myofibers are responsible for contraction through the sequential activation of sarcomeres arranged in series and account for approximately 80% of the cytoplasm volume [2]. In addition to myofibers, skeletal muscle contains several other critical cell populations. Satellite cells, the resident stem cells, are essential for muscle regeneration. Resident macrophages contribute to immune surveillance and tissue repair [3], while endothelial cells support vascular function and nutrient delivery. Fibro-adipogenic progenitor cells (FAPs) are also present and possess multipotent differentiation potential, giving rise to adipocytes, fibroblasts, and, under specific conditions, chondrocytes or osteoblasts [4]. In response to trauma, the inflammatory cascade is typically triggered by sentinel cells, primarily resident mast cells and macrophages in skeletal muscle that detect tissue damage. Upon injury recognition, resident macrophages differentiate from their M0 state (immature monocytes) into the pro-inflammatory M1 phenotype. These M1 macrophages secrete cytokines such as interleukin-1 (IL-1), IL-6 and tumour necrosis factor alpha (TNF-α), which amplify inflammation and contribute to tissue degradation [5,6]. Concurrently, angiogenesis and vasodilatation enhance the recruitment of circulating immune cells, including neutrophils, monocytes and lymphocytes, to the injury site. Chemokines such as C-X-C motif chemokine ligand 8 (CXCL8/IL-8), chemokine C-C motif ligand/monocyte chemoattractant protein-1 (CCL2/MCP-1) and (RANTES/CCL5) mediate the chemotactic migration of these leukocytes [5,7,8]. These immune cells play a critical role in clearing cellular and extracellular matrix (ECM) debris and regulating oxidative stress. As part of the tissue repair process, the inflammatory response resolves through a phenotypic shift in macrophages from the M1 to the anti-inflammatory M2 state. M2 macrophages release cytokines such as IL-10 and transforming growth factor beta (TGF-β), which promote muscle regeneration and ECM remodelling [5]. During the regenerative phase, FAPs contribute to the ECM remodelling, while satellite cells exit quiescence, proliferate, differentiate into myotubes, and fuse with damaged myofibers. This coordinated process not only facilitates effective tissue repair but also helps the maintenance of the satellite cell pool, ensuring long-term regenerative capacity [9].

Muscle regeneration is not always fully restorative. In certain cases, repeated cycles of degeneration/regeneration can result in fibrosis, characterized by excessive ECM deposition and impaired muscle function. Additionally, Ca^2+^ dysregulation is implicated in various pathological conditions, including soft tissue calcification and, in more severe instances, heterotopic ossification (HO), the abnormal formation of bone within soft tissues. This review provides a comprehensive overview of the current understanding of the mechanisms driving HO in skeletal muscle. We examine the signalling pathways that mediate macrophage phenotypic transitions during inflammation, the development of an abnormal ECM, and the aberrant differentiation of FAPs into chondrocytes or osteoblasts. We also explore the role of neuro-endocrine signalling in the pathogenesis of neurogenic heterotopic ossification (NHO). Finally, we review existing therapeutic strategies for HO and discuss emerging approaches that hold promise for more effective prevention and treatment.

## 2. Heterotopic Ossification

HO refers to the abnormal formation of bone in non-skeletal tissues such as skeletal muscle. It is commonly observed following severe burns, multiple traumas, orthopedic surgeries or in genetic disorders like *fibrodysplasia ossificans progressiva* (FOP) [10]. The process of ectopic bone formation after a muscle injury begins with mineralization of inorganic ions, followed by the recruitment of immune cells like macrophages and the presence of osteogenic factors leading to the differentiation of macrophages into a pro-osteogenic phenotype and the differentiation of FAPs into chondrocytes and subsequently osteoblasts [11]. Macrophages can also promote FAP differentiation by secreting osteogenic factors. This cascade ultimately leads to *myositis ossificans*, the development of HO within skeletal muscle tissue. However, this proposed cascade of events underlying HO remains speculative due to the absence of detailed studies mapping its chronological progression.

While ectopic inorganic calcium phosphate deposits (EDs) and HO both involve mineral accumulation in soft tissues, they represent distinct pathological processes. Both are characterized by the presence of hydroxyapatite Ca_5_(PO_4_)_3_OH, the primary mineral component of normal bone [12,13]. However, the key difference lies in their underlying mechanisms: HO involves cellular reprogramming toward an osteogenic lineage [14], whereas ED occurs without the involvement of progenitor cells, resulting from passive mineral deposition. Thus, while both conditions fall under the broader category of ectopic mineralization, only HO is associated with cell-mediated ossification in soft tissues.

## 3. Causes of Heterotopic Ossification: Genetic and Trauma-Related Etiologies

### 3.1. Genetic

HO can arise from multiple etiologies, including acquired and genetic causes (Figure 1). Among the genetic forms, FOP is the most extensively characterized. This rare disease is caused by a gain-of-function mutation in the activin A receptor type I/activin receptor-like kinase 2 (*ACVR1/ALK2*)—most commonly the ALK2^R206H^ variant, which leads to constitutive activation of the bone morphogenetic protein (BMP) signalling pathway [15,16]. FOP affects 1 in 2 million individuals, with no known sex predilection [17]. Other related genetic disorders include Albright hereditary osteodystrophy (AHO) and progressive osseous heteroplasia (POH), both of which result from inactivating mutations in the guanine nucleotide-binding protein, alpha stimulating activity polypeptide (*GNAS*). This gene encodes the α-subunit of the stimulatory G protein (Gαs) [18,19], which regulates several intracellular signalling pathways, including parathyroid hormone (PTH) signalling [20]. PTH signalling plays a critical role in maintaining Ca^2+^ and PO_4_^3−^ homeostasis. However, epidemiological data on the prevalence of AHO and POH are still limited.

### 3.2. Trauma-Induced HO

The most common cause of HO is trauma or injury/damage typically resulting from accident or invasive surgical procedures (Figure 1). Traumatic HO can be broadly categorized into two types: non-neurologic trauma, which includes orthopedic or soft tissue injuries, and neurologic trauma, such as spinal cord or brain injuries (Figure 1). Among non-neurologic causes, surgical interventions, particularly joint replacement surgeries, are well-documented to trigger HO. For instance, Toom et al. (2001) reported that 32% of patients who underwent total hip arthroplasty developed HO within a year post-surgery [21]. Similarly, HO is frequently observed following shoulder surgeries, with incidence rates ranging between 6.6% and 54% [22]. Comparable occurrences have also been documented after knee surgeries [23]. Given the high volume of joint replacement procedures and the significant risk of HO development following these interventions, the condition can create a considerable societal burden. As of 2010, approximately 7 million individuals in the United States were living with hip and knee replacements [24].

Thermal injuries are another recognized cause of HO, with over 20% of burn patients developing ectopic bone formation [25]. Additionally, HO can be triggered by repetitive trauma, such as recurrent contusions or micro-injuries, which are common in contact sports like rugby, football, and martial arts [26]. While not all traumatic lesions lead to HO, the risk following quadriceps trauma is estimated to be between 9% and 20% [27]. Inadequate recovery time may exacerbate this risk by contributing to a dysregulated inflammatory response [28], potentially facilitating HO development after sport-related injury [26]. Bone fractures are another significant contributor to HO in both civilian and military populations [29]. In civilians, HO occurs in 20% of forearm fractures and up to 52% of femoral shaft fractures [29]. Among military personnel, the incidence is even higher, with HO observed in over 65% of cases following blast injuries or amputations [30], likely due to complex polytrauma involving bone, muscle and nerve injuries [30].

The second major category of trauma-induced HO arises from neurological injuries, particularly spinal cord injury (SCI) and traumatic brain injury (TBI) (Figure 1). In cases of SCI, approximately 20% of patients develop NHO [31], with ectopic bone formation consistently occurring below the level of the spinal lesion [32]. Interestingly, the absence of NHO in conditions such as spinal muscular atrophy (SMA), a genetic disorder characterized by motoneuron (MN) degeneration, suggests that MN loss alone is insufficient to trigger NHO. Instead, a combination of phenotypic alterations in MNs and neuroimmune signalling likely contributes to the pathogenesis of NHO. TBI is also a significant risk factor for NHO. While the overall incidence of NHO in TBI patients is around 11% [33], the prevalence can escalate dramatically to as high as 90% when TBI is accompanied by an elbow injury [33], compared to only 3–6% in patients with an elbow injury alone [34]. Animal studies further support this association: mice subjected to TBI and subsequent hip surgery developed significantly more NHO compared to those without brain injury [35]. These findings underscore the role of neurological trauma in exacerbating HO formation. The mechanisms underlying SCI- and TBI-induced HO will be further explored in great detail in the section on neuroendocrine signalling.

## 4. Abnormal Inflammation in Heterotopic Ossification

The inflammatory response in HO becomes dysregulated, leading to abnormal cellular phenotypes and aberrant ECM remodelling [36]. Using a parabiosis model with green fluorescent protein (GFP)-positive/luciferase-expressing donor mice, it has been demonstrated that circulating immune cells are prominently involved during the inflammatory and proliferative phases of HO, with a reduced presence in mature lesions. This was observed following Achilles tendon transection combined with dorsal burn injury to induce HO [37]. Building on these findings, single-cell RNA sequencing, spatial transcriptomics and lineage tracing analyses have revealed that neutrophils, natural killer cells (NKs), macrophages, cluster differentiation 4 (CD4+) and CD8+ T lymphocytes can promote chondrogenic and/or osteogenic mesenchymal stem cell (MSC) lineages [38]. Pharmacological reduction in neutrophils at the injury site using agents such as hydroxychloroquine or toll-like receptor 9 inhibitors has been shown to significantly reduce HO formation [39]. Moreover, pro-inflammatory M1 macrophages are particularly abundant during the early phases of HO [40,41], underscoring their central role in initiating the ossification cascade. M1 macrophages persist in the lesion environment, secreting cytokines that not only sustain inflammation but also promote chondrocyte and osteogenic differentiation. Depletion of macrophages using clodronate liposomes significantly reduced ectopic bone formation [41]. One of the key mediators secreted by M1 macrophages is oncostatin M (OSM), which has been shown to promote HO (Figure 2) by activating the Janus kinase 2/Signal transducer and activator of transcription 3 (JAK2/STAT3) signalling pathway in ligamentum flavum cells [42,43]. Additionally, IL-1b, secreted by M1 macrophages, activates the nuclear factor-kappa B (NFκB) pathway [44] and is involved in the development of HO [45,46]. While M2 macrophages are generally associated with the regenerative phase of tissue repair, their role in HO remains controversial. Recent studies suggest that M2 macrophages may actively contribute to HO development. Using an Achilles tendon injury model and an in vitro collagen I calcification system, Han et al. (2023) demonstrated that macrophage-derived extracellular deoxyribonucleic acid (DNA), particularly from M2 macrophages, is involved in pathological calcification [47]. Consistent with these findings, Tu et al. (2023) reported that macrophage depletion significantly suppressed traumatic HO formation in a murine model [48], further indicating M2 macrophage contribution to HO progression [38]. In contrast, promoting M2 polarization has been shown to attenuate HO [49]. For instance, inhibition of focal adhesion kinase-2 (FAK2), which drives macrophages toward an M2 phenotype, was found to significantly reduce HO incidence [49] (Figure 2). These findings highlight the complex and potentially dual role of M2 macrophages in HO pathogenesis, suggesting that while they contribute to tissue repair, they may also facilitate HO under certain conditions. 

TGF-β1 plays a multifaceted role in the pathogenesis of HO, influencing both chondrogenic differentiation and macrophage function [50] (Figure 2). During the inflammatory phase, TGF-β1 is secreted by macrophages and can act in an autocrine manner via ALK5 signalling, potentially promoting HO [51]. Another critical mediator in HO development is activin A (Figure 2), a member of the TGF-β superfamily, produced by a variety of cell types such as dendritic cells, macrophages and fibroblasts. Notably, antibody-mediated neutralization of activin A significantly reduces HO formation, confirming its role in disease progression [52]. Cytokines from the BMP family, particularly BMP2, BMP4 and BMP7, also play central roles in HO development [45,53,54] (Figure 2). These ligands signal through BMP type I and type II receptors, activating downstream the BMP/small mother against decapentaplegic (BMP/SMAD) (Figure 2) or p38/mitogen-activated protein kinases (p38/MAPK) pathways (Figure 2) [55,56], which, in turn, induce the expression of osteogenic markers such as Runt-related transcription factor 2 (RUNX2) or osterix [57,58] (Figure 2). Furthermore, BMP6 has also been shown to influence macrophage transcriptomics [59] and promote osteogenic differentiation as evidenced by increased alkaline phosphatase expression in C2C12 muscle cells [60,61]. Furthermore, implantation by intramuscular injection of C2C12 myoblasts overexpressing BMP6 in the quadriceps of mice clearly shows development of HO [61]. However, serum BMP6 levels in patients with FOP were not significantly elevated compared to the control group [62,63], suggesting a limited systemic role. Nonetheless, BMP6 may contribute to the early stage of HO by promoting chondrocyte differentiation and osteoblast lineage in skeletal muscles (Figure 2). Altogether, these findings underscore the role of cytokines and growth factors in shaping the inflammatory and osteogenic microenvironment that drives HO development.

The involvement of B and T lymphocytes in HO remains a subject of debate. Ranganathan et al. (2016) showed a reduced HO in recombination-activating gene 1 (Rag-1) knockout mice, which lack mature lymphocytes [64]. In contrast, Alexander et al. (2022), using a different HO model, observed no such reduction [65]. The discrepancy may stem from different HO models: Ranganathan et al. (2016) used a burn and Achilles tenotomy model [64], while Alexander et al. (2022) employed SCI combined with a cardiotoxin (CTX) injection [65]. Additional studies suggest that lymphatic vessels may contribute to HO formation, particularly in burn and tenotomy models [66,67]. For instance, lymph node excision has been shown to reduce HO formation, supporting the involvement of lymphatic tissue [65]. Conversely, Zhang et al. (2021) found that inhibition of lymphatic endothelial cells (LECs) exacerbated HO [66]. Their study demonstrated that fibroblast growth factor receptor 3 (FGFR3) signalling plays a regulatory role in HO, with FGFR3 inhibition promoting osteogenic differentiation, which is coherent with previous findings [68]. Thus, the precise role of lymphocytes and lymphatic tissue in HO remains unclear. Current evidence suggests that lymphocytes may contribute to HO in trauma-induced models such as tenotomy and burn injury. The following section will explore the abnormalities in ECM composition and remodelling associated with HO.

## 5. An Aberrant New Extracellular Matrix

The composition and structure of the ECM, primarily secreted by fibroblasts, play a pivotal role in the pathogenesis of HO. Fibroblasts contribute to this process by recruiting leukocytes, thereby converting acute inflammation into a chronic state that promotes fibrosis and impairs normal tissue repair [4]. Under the influence of mechanical cues and signalling pathways, including BMP, TGF-β and Wnt/β-catenin, fibroblasts can transdifferentiate into osteoblast-like cells, further driving HO formation [68,69]. Following traumatic injury, the resulting tissue damage, inflammation, fibrosis and localized hypoxia create a permissive microenvironment that activates the hypoxia-inducible factor 1-alpha/procollagen-Lysine, 2-Oxoglutarate 5-Dioxygenase 2 (HIF1-α/PLOD2) axis [70] (Figure 2). This pathway is central to ECM remodelling and promotes a pro-osteogenic phenotype in MSCs [70,71]. Conversely, conditional deletion of HIF1-α in MSCs significantly reduces HO formation in a model of burn/tendon injury [69]; MSCs contribute to ECM remodelling both indirectly through differentiation into osteoblasts and directly by secreting metalloproteinases (MMPs) and ECM proteins [72]. Upon osteogenic commitment, MSCs generate an aberrant ECM with a high propensity for mineralization. Additionally, fibroblasts expressing HIF1-α impair muscle regeneration and disrupt collagen cross-linking, further facilitating the onset of HO [70]. The newly formed ECM in this context is often acellular [73], a characteristic that enables it to chelate BMP 2, thereby promoting the differentiation of osteoblasts and osteoclasts [73]. This remodelling process includes the deposition of structural proteins such as type II collagen and the expression of MMPs, like MMP9, which serve as a biomarker of HO in murine models [74]. Furthermore, ECM remodelling plays a critical role in determining stem cell fate. Elevated collagen concentrations within the ECM have been shown to enhance its mineralization potential, thereby promoting osteogenesis [75]. This reciprocal interaction establishes a positive feedback loop that amplifies and sustains HO. Interestingly, while ECM remodelling contributes to HO progression, mechanical factors can also modulate this process. Several studies have also reported that mechanical unloading and joint immobilization serve as an effective conservative treatment to reduce HO. Immobilization attenuates mechanotransduction signalling, shifting MSCs differentiation toward adipogenesis rather than osteogenesis, ultimately decreasing HO formation [76]. Conversely, mobilization and mechanical loading stimulate MSC differentiation into chondrocytes and osteoblasts, particularly in the context of a hypoxic and inflammatory microenvironment, thereby promoting HO development [76]. Among the MSC populations implicated in this mechanosensitive process, FAPs play a central role and will be discussed in the following section.

## 6. FAP at the Core of Myositis Ossificans

FAPs are a specialized subset of MSCs residing in skeletal muscles, where they play a critical role in maintaining the ECM and supporting muscle regeneration. Under normal physiological conditions, FAPs differentiate into fibroblasts or adipocytes to facilitate tissue repair. However, in pathological contexts, such as trauma or chronic inflammation, FAPs may aberrantly differentiate into osteoblasts, contributing to ectopic bone formation, a defining feature of myositis ossificans.

In FOP, FAPs have been directly implicated due to a gain-of-function mutation in the *ACVR1* gene [71]. This mutation, most commonly ACVR1 (R206H), alters the receptor’s response to ligands, enabling inappropriate osteogenic signalling (Figure 2). The activin receptor is a heteromeric complex composed of two type I and two type II receptors [77]. Type II receptors bind ligands and facilitate the phosphorylation and activation of type I receptors, which then initiate intracellular signalling cascades. Under normal conditions in muscle fibres, activin signalling predominantly activates the Smad2/3/4 pathway, promoting catabolic processes and inhibiting muscle growth [77]. In FOP, however, the ACVR1 (R206H) mutation causes the receptor to respond abnormally to activin A by activating the Smad1/5/8 pathway [78], typically reserved for BMP signalling (Figure 2). This aberrant activation drives the osteogenic differentiation of FAPs [71].

FAP also expresses receptors for TGF-β, yet the role of TGF-β signalling in MSC differentiation and bone metabolism remains controversial. Wei et al. (2024) highlighted the divergent perspectives on TGF-β’s influence on osteogenic differentiation in bone-derived MSCs [79]. Importantly, FAPs can adopt an osteogenic or chondrogenic phenotype in response to BMP signalling [80,81] (Figure 2). For instance, exposure to BMP-2 in an inflammatory microenvironment has been shown to induce osteogenic differentiation in FAPs [80]. FAPs are a major cellular source of HO lesions, with their osteogenic differentiation primarily driven by an altered inflammatory microenvironment [80]. Within this environment, activin A, TGF-β1, and BMPs, all members of the TGF-β superfamily, play pivotal roles in promoting pathological signalling. Thus, targeting inflammation may represent an effective strategy to prevent the aberrant differentiation of FAPs into osteogenic or chondrogenic lineages.

Beyond cytokine and growth factor signalling, the mechanotransduction pathway also contributes to FAP fate decisions. The transcriptional coactivators Yes-associated protein (YAP) and transcriptional coactivator with PDZ-binding motif (TAZ) are key regulators of MSC osteogenesis [82] (Figure 2). YAP/TAZ reside in the cytoplasm and translocate to the nucleus in response to ECM remodelling or by inactivation of the Hippo pathway [83]. Once in the nucleus, they activate gene expression programmes associated with survival, proliferation and osteogenesis, including the master osteogenic regulator Runx2 [84,85]. ECM stiffness enhances YAP/TAZ nuclear translocation and, when combined with BMP2 signalling, promotes osteogenic differentiation in C2C12 myoblasts [86]. High ECM stiffness inactivates the Hippo pathway, allowing YAP/TAZ to accumulate in the nucleus [86]. Conversely, active Hippo signalling retains YAP/TAZ in the cytoplasm, preventing their transcriptional activity. Discoidin domain receptor 2 (DDR2), a transmembrane receptor for type I collagen, further facilitates YAP/TAZ nuclear translocation [87] (Figure 2). DDR2 binding to collagen inhibits YAP phosphorylation at serine 127 and suppresses large tumour suppressor kinases 1 and 2 (LATS1/2) activity, thereby preventing YAP/TAZ degradation [82]. Given that type I collagen is a hallmark of HO, DDR2-mediated YAP/TAZ activation likely contributes to the osteogenic differentiation of FAPs.

## 7. Neuroendocrine Signalling in Neuro-Heterotopic Ossification

As discussed in previous sections, SCI or TBI markedly increases the risk of developing HO [33]. Both SCI and TBI are associated with neuropathic pain, which is mediated by nerve growth factor (NGF). NGF, secreted by vascular smooth muscle cells and pericytes at the site of injury, binds to its high-affinity receptor tropomyosin receptor kinase A (TrkA) on sensory neurons. This interaction activates the transient receptor potential vanilloid-1 (TRPV1) channel, triggering a signalling cascade that includes the retrograde transport of the NGF-TrkA-TRPV1 complex to the neuronal cell body [88]. This process triggers the nociceptor-induced neural inflammation, a key driver of NHO. Neural inflammation also leads to the release of neuropeptides such as substance P (SP), calcitonin gene-related peptide (CGRP) and OSM. These molecules activate multiple signalling pathways that promote osteogenesis [89]. SP, a key neuropeptide released in response to tissue damage, binds to neurokinin-1 (NK-1) receptors on target cells, initiating neurogenic inflammation through phospholipase C and adenylate cyclase signalling cascades [90]. These pathways subsequently activate the mechanistic target of rapamycin (mTOR) and NFκ-B pathways, both of which are implicated in inflammatory and osteogenic responses. In a combined SCI and CTX injury model, elevated circulating levels of SP were correlated with increased HO volume [91]. Notably, the pharmacological inhibition of SP signalling using an NK-1 receptor antagonist resulted in a 30% reduction in HO volume [91], highlighting SP-NK-1 signalling as a potential therapeutic target.

In the neuroinflammatory environment, SP acts synergistically with CGRP to amplify pro-osteogenic signalling [89]. CGRP synthesis in sensory neurons is NGF-dependent, stored in sensory nerve terminals and released in response to nociceptive stimuli [89]. In the context of TBI-induced HO, CGRP has been implicated in immune cell recruitment, and the osteogenic differentiation of myeloid progenitor cells (MPCs) through activation of the adenylate cyclase-cAMP/protein kinase A-C-AMP response element-binding protein (adenylate cyclase-cAMP/PKA-CREB) signalling cascade [89]. Notably, exogenous CGRP administration in mice subjected to the combined SCI + CTX injury model led to chondrocyte differentiation and spontaneous HO formation [92]. However, in a tendinopathy-induced HO model, co-administration of CGRP and SP paradoxically prevented HO development [93]. These findings underscore the context-dependent and potentially opposing roles of SP and CGRP in NHO, highlighting the complexity of their interactions and downstream signalling networks. Moreover, OSM is highly secreted after a spinal cord injury and leads to the osteogenic differentiation of ligamentum flavum cells [94]. Despite growing evidence implicating the central nervous system (CNS) in HO pathogenesis, the precise molecular mechanisms remain incompletely understood. SP and CGRP are central mediators of neuroinflammation and may coordinate the systemic inflammation response following musculoskeletal trauma, contributing to the initiation and progression of ectopic bone formation.

## 8. Current and Potential Treatments for Heterotopic Ossification

Currently, there is no definitive cure for HO. Existing treatments are limited in efficacy, often fail to fully prevent HO, and are frequently associated with significant side effects. Radiation therapy is one approach used to inhibit osteochondral differentiation of resident progenitor cells such as FAPs or MPCs by inducing DNA damage [95]. However, repeated and high doses of radiation can lead to cell death and increase the risk of malignancy and may not be well tolerated by all patients due to long-term adverse effects [95]. Nonsteroidal anti-inflammatory drugs (NSAIDs) are commonly used to manage the inflammatory phase of HO and prevent its dysregulation. While NSAIDs can be effective, prolonged use is associated with significant side effects, including gastrointestinal bleeding, renal impairment, cardiovascular events, and CNS complications [96]. Excision of ectopic bone is another option, but it requires the lesion to be sufficiently mature and well-demarcated. Moreover, surgery itself can trigger a renewed inflammatory response, increasing the risk of HO recurrence [95]. To date, there is no strong evidence supporting the efficacy of physical therapy in improving the conditions of patients with HO [97]. Given the limitations of current therapies, there is an urgent need for novel and more effective therapeutic strategies. A recent study has shown that SCI triggers a rapid spike in corticosterone levels and that administration of corticosterone or the synthetic glucocorticoid receptor (GR) agonist, dexamethasone, is sufficient to induce HO in skeletal muscle even in the absence of SCI [98]. Conversely, the use of GR antagonists or selective deletion of the GR gene has been demonstrated to significantly reduce HO volume, highlighting the glucocorticoid signalling pathway as a promising therapeutic target [98]. Finally, modulating calcium-regulating pathways such as those involving parathyroid hormone [99,100], osteoprotegerin (OPG) [101,102], ectonucleotide pyrophosphate/phosphodiesterase 1 (ENPP1) [103,104] or P2X7 purinergic receptor [105,106,107] may offer alternative preventive strategies by influencing mineralization and osteogenic signalling.

## 9. Conclusions

HO is a pathological condition characterized by the abnormal formation of bone within soft tissues, including skeletal muscles. HO can arise from genetic mutation, as seen in FOP, or be acquired following trauma. In trauma-induced HO, injuries to the CNS, such as TBI or SCI, significantly increase the risk of ectopic bone formation. Following trauma, an uncontrolled inflammatory response coupled with the release of pro-osteogenic cytokines such as BMPs and TGF-β drives the aberrant differentiation of resident progenitor cells such as FAPs and MPCs. This process contributes to the pathological calcification of soft tissues. Moreover, the neuroinflammatory environment following CNS trauma exposes FAPs to neuropeptides such as SP and CGRP, which further promote their osteogenic transition. Currently, there are no efficient treatments for HO. Available therapeutic approaches offer limited efficacy and are often associated with significant risks and adverse effects. Although HO is a well-characterized condition requiring osteogenic progenitor cells, specific molecular signals and a permissive environment to initiate and sustain ectopic bone formation, its underlying molecular mechanisms remain incompletely understood. Continued research is essential to elucidate these pathways and to develop safe, targeted, and effective therapeutic strategies.

## Figures and Tables

**Figure 1 ijms-26-07516-f001:**
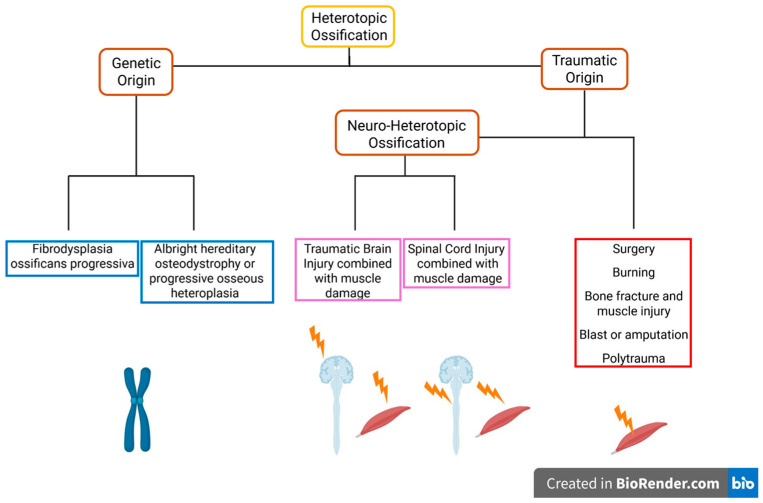
Pathogenesis of heterotopic ossification. Pathological bone formation within soft tissues, such as skeletal muscle, can arise from genetic disorders or in response to trauma. Ectopic ossification frequently follows surgical procedures, thermal injuries, or repetitive microtrauma from contact sports. Moreover, the simultaneous injury to the nervous system and skeletal muscle markedly increases the risk of developing heterotopic ossification. This image was created with BioRender (https://biorender.com/, accessed on 9 June 2025).

**Figure 2 ijms-26-07516-f002:**
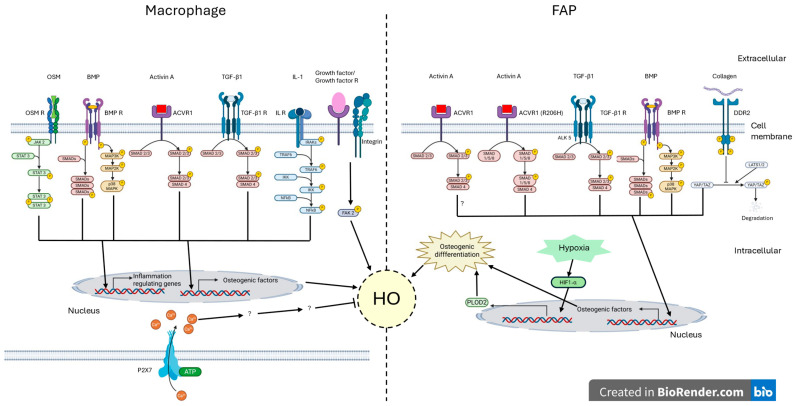
Macrophage and fibro-adipogenic progenitor reprogramming in the pathogenesis of heterotopic ossification. Macrophages and fibro-adipogenic progenitors (FAPs) are key contributors to the pathogenesis of heterotopic calcification. Through various signalling pathways, macrophages secrete osteogenic factors that drive the osteogenic differentiation of FAPs. This interaction plays a pivotal role in the development and progression of heterotopic ossification. This image was created with BioRender (https://biorender.com/, accessed on 9 June 2025). Abbreviations: ACVR1: Activin A Receptor Type I; BMP: Bone Morphogenetic Protein; BMP R: Bone Morphogenetic Protein Receptor; DDR2: Discoidin Domain Receptor 2; FAK 2: Focal Adhesion Kinase 2; Growth factor R: Growth factor Receptor; HIF1-α: Hypoxia-Inducible Factor 1-alpha; IKK: IkappaB kinase; IL-1: Interleukin 1; IL R: Interleukin Receptor; IRAKs: Interleukin-1 Receptor-Associated Kinases; JAK 2: Janus Kinase 2; LATS1/2: Large Tumour Suppressor Kinases 1 and 2; MAP2K: Mitogen-Activated Protein Kinase Kinases; MAP3K: Mitogen-Activated Protein Kinase Kinase Kinases; NFκB: Nuclear Factor-Kappa B; OSM: Oncostatin M; OSM R: Oncostatin M Receptor; p38: p38 mitogen-activated protein kinases; PLOD2: Procollagen-Lysine, 2-Oxoglutarate 5-Dioxygenase 2; SMADs: Small Mother Against Decapentaplegic; SMAD1/5/8: Small Mother Against Decapentaplegic 1/5/8; SMAD2/3: Small Mother Against Decapentaplegic 2/3; SMAD4: Small Mother Against Decapentaplegic 4; STAT 3: Signal Transducer and Activator of Transcription 3; TGF-β1: Transforming Growth Factor Beta 1; TGF-β1 R: Transforming Growth Factor Beta 1 Receptor; TRAF6: Tumor Necrosis Factor Receptor-Associated Factor 6; YAP/TAZ: Yes-associated protein/transcriptional coactivator with PDZ-binding motif.

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
