# Peer review of "When Bone Forms Where It Shouldn’t: Heterotopic Ossification in Muscle Injury and Disease"

_ijms, 2025, doi:10.3390/ijms26157516_

Round 1
Reviewer 1 Report
Comments and Suggestions for Authors
This is a very interesting review article, that clearly lines out heterotopic ossification in detail and what happens when it forms in unexpected places. However, the reviewers focus on the immunsystem and less on the process of differentiation that happens to the cell types responsible for the ossification. The reviewers should include more detail on mechanistic studies, what receptors, the role of different signaling pathways etc. While the review starts out strong the important mechanisms are not fully discussed
Author Response
Dear reviewers,
We sincerely thank you for your thoughtful and constructive comments as well as the time you dedicated to enhancing the quality of our manuscript. We deeply appreciate your insightful suggestions and are pleased to address them in detail.
Reviewer 1:
“However, the reviewers focus on the immune system and less on the process of differentiation that happens to the cell types responsible for the ossification. The reviewers should include more detail on mechanistic studies, what receptors, the role of different signalling pathways etc. While the review starts out strong the important mechanisms are not fully discussed.”
Although the differentiation of fibro-adipogenic progenitors (FAPs) into osteogenic cells represents a pivotal step in the development of heterotopic ossification (HO) within skeletal muscle, the molecular mechanisms governing this process remain incompletely understood, particularly in this specific tissue context. In our review, we have intentionally focused on the inflammatory processes that initiate HO in skeletal muscle, reflecting the core expertise of our research group. Nonetheless, we have included references to key signalling pathways implicated in FAP differentiation, such as BMP, Wnt/β-catenin, and TGF-β, which are briefly discussed within the broader framework of inflammation-mediated signalling.
Our objective was not to provide an exhaustive overview of osteogenic signalling pathways, but rather to emphasize the role of inflammation in orchestrating the early phases of HO and its potential to prime FAPs for transdifferentiation. For more in deepth discussions of individual pathways, we refer readers to recent focused reviews such as Zhao et al. (2024) on Wnt/β-catenin signaling [1] and Wei et al. (2024) on the dual role of TGF-β in osteogenic differentiation [2]. We believe this approach allows us to maintain a coherent and focused scope while acknowledging the broader complexity of the field.
Reviewer 2:
“Consider clarifying the sequence of events leading from macrophage activity to FAP differentiation, as this is a central mechanistic point;”
Regarding the sequence of events leading to HO in skeletal muscle, we have revised the relevant sections of the manuscript to enhance clarity and readability. Additionally, we have modified the graphical abstract to offer a more intuitive overview of the complex interplay among the proteins and cellular events discussed throughout the review. Given that the pathophysiological cascade underlying HO in skeletal muscle remains poorly defined, the sequence we present reflects the most plausible model based on the current literature, while acknowledging that definitive experimental validation is still lacking.
“Expand slightly on the neuroendocrine mechanisms in NHO; readers may benefit from more detail or a cited example.”
We acknowledge the growing importance of neuroendocrine signalling in the development of HO in skeletal muscle, particularly following central nervous system injury, an area that is rapidly emerging in the field. In response to the reviewer's comment, we have added oncostatin M to the relevant section. This cytokine, which is upregulated after spinal cord injury, has been implicated in musculoskeletal remodelling and play a role in HO pathogenesis.
Once again, we thank the reviewers for their thoughtful feedback, which has significantly contributed to improving both the clarity and scientific depth of our manuscript.
References
[1] Y. Zhao, F. Liu, Y. Pei, F. Lian, and H. Lin, ‘Involvement of the Wnt/β‐catenin signalling pathway in heterotopic ossification and ossification‐related diseases’, J. Cell. Mol. Med., vol. 28, no. 18, p. e70113, Sep. 2024, doi: 10.1111/jcmm.70113.
[2] E. Wei et al., ‘TGF-β signaling regulates differentiation of MSCs in bone metabolism: disputes among viewpoints’, Stem Cell Res. Ther., vol. 15, no. 1, p. 156, May 2024, doi: 10.1186/s13287-024-03761-w.
Reviewer 2 Report
Comments and Suggestions for Authors
This review explores cellular and molecular mechanisms of HO, emphasizing the roles of inflammation, trauma, genetic disorders and neurologic injuries. The topic is relevant, it opened up a whole new word of knowledge about this pathology and it is very well-written. Each mechanism associated with HO is treated in depth with the support of scientific literature. I only suggest some minor revisions:
- Consider clarifying the sequence of events leading from macrophage activity to FAP differentiation, as this is a central mechanistic point;
- Expand slightly on the neuroendocrine mechanisms in NHO; readers may benefit from more detail or a cited example.
To conclude, this is a strong and valuable review that addresses an important area of translational research and, thanks to this consideration, could be accepted after these minor revisions. Congrats to the authors!
Author Response

(The authors gave the same response as above.)
